# OpenFOAM Simulations of Late Stage Container Draining in Microgravity

**Joshua McCraney** [1,*,†], **Mark Weislogel** [2] **and Paul Steen** [1,3]

1   School of Mechanical and Aerospace Engineering, Cornell University, Ithaca, NY 14853, USA; phs7@cornell.edu
2   School of Mechanical Engineering, Portland State University, Portland, OR 97207, USA; weisloge@pdx.edu
3   School of Chemical and Biomolecular Engineering, Cornell University, Ithaca, NY 14853, USA
*   Correspondence: jm2555@cornell.edu
†   This paper is an extended version of our paper published in 15th OpenFOAM Workshop.

**Abstract:** In the reduced acceleration environment aboard orbiting spacecraft, capillary forces are often exploited to access and control the location and stability of fuels, propellants, coolants, and biological liquids in containers (tanks) for life support. To access the 'far reaches' of such tanks, the passive capillary pumping mechanism of interior corner networks can be employed to achieve high levels of draining. With knowledge of maximal corner drain rates, gas ingestion can be avoided and accurate drain transients predicted. In this paper, we benchmark a numerical method for the symmetric draining of capillary liquids in simple interior corners. The free surface is modeled through a volume of fluid (VOF) algorithm via interFoam, a native OpenFOAM solver. The simulations are compared with rare space experiments conducted on the International Space Station. The results are also buttressed by simplified analytical predictions where practicable. The fact that the numerical model does well in all cases is encouraging for further spacecraft tank draining applications of significantly increased geometric complexity and fluid inertia.

**Keywords:** capillary flow; container draining; CFD; interior corner; capillary fluidics; contact angle; tankage

## 1. Introduction

Capillary draining dictates the fluid withdrawal rate of precious fuels, propellants, coolants, and aqueous solutions prevalent on spacecraft. For the specific example of liquid fuels aboard orbiting spacecraft, capillary draining can serve as a limit to the life of the spacecraft if and when the residual fuel in the tank is or becomes inaccessible. It is essential to establish the maximum capillary flow rate at which a container in a microgravity environment can be drained. Interior corner devices constructed of 'vanes' provide a geometric family of propellant management devices (PMDs) [1]. Such constructs have also been studied for the practical purposes of enhancing bubble coalescence and/or breakup [2,3], stabilization of liquid columns from 'g-jitter' induced by orbital maneuvers, docking and crew activity [4], and inhibiting choked flows [5]. Interior corners can be used to both enhance and hinder flows [6]. Herein, we study simple interior linear corners and corners with increased geometric complexity.

A drain application is sketched in Figure 1, where a right circular cylinder is drained in low-g. As liquid drains, its topology changes, and a dry region forms at the base. It is this late stage draining (Figure 1, far right) we consider, as such transients can be critical for fluid withdrawal. For sufficiently small stream-wise curvature, one can imagine a 'cut' along the single drain port, the unraveling of

which yields a linear corner (cross section shown bottom right of Figure 1) with a drain at each end: hereafter referred to as a double-drain (Figure 2). In this work, we simulate double-drain problems. We quantitatively assess meniscus evolution profiles, maximal interfacial height, and volumetric drain rates. Simulations are validated against a simplified analytical model formulated and experimentally analyzed by Weislogel and McCraney [7] (ICF-1 test cell below). The simulations are then extended to model new, more complex geometries (ICF-8 test cell below) also conducted experimentally aboard the International Space Station (ISS).

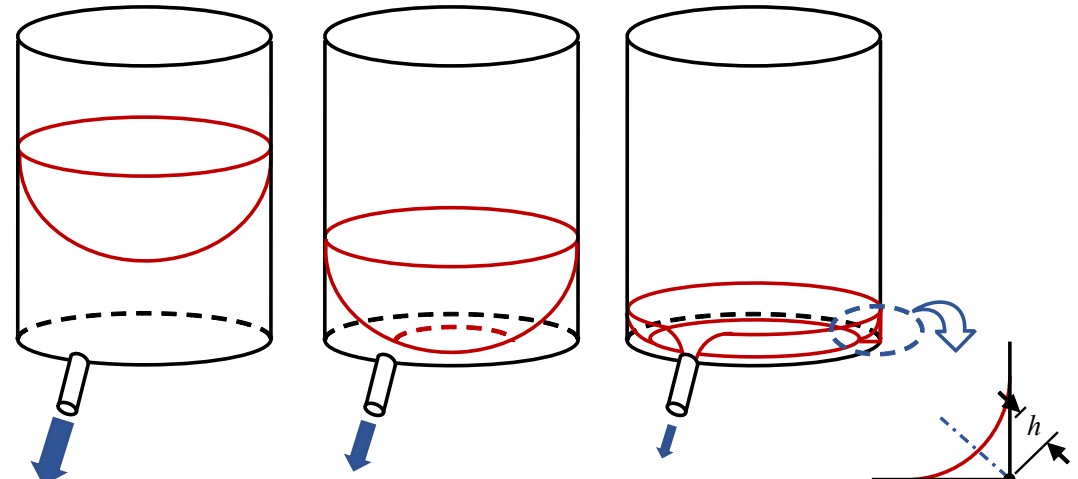

**Figure 1.** Simple example of capillary draining in a zero-gravity environment. The late-stage draining condition (**right**) yields a thin visco-capillary flow that is approximately linear, with approximately 2D-Cartesian cross-section sketched (**far right**).

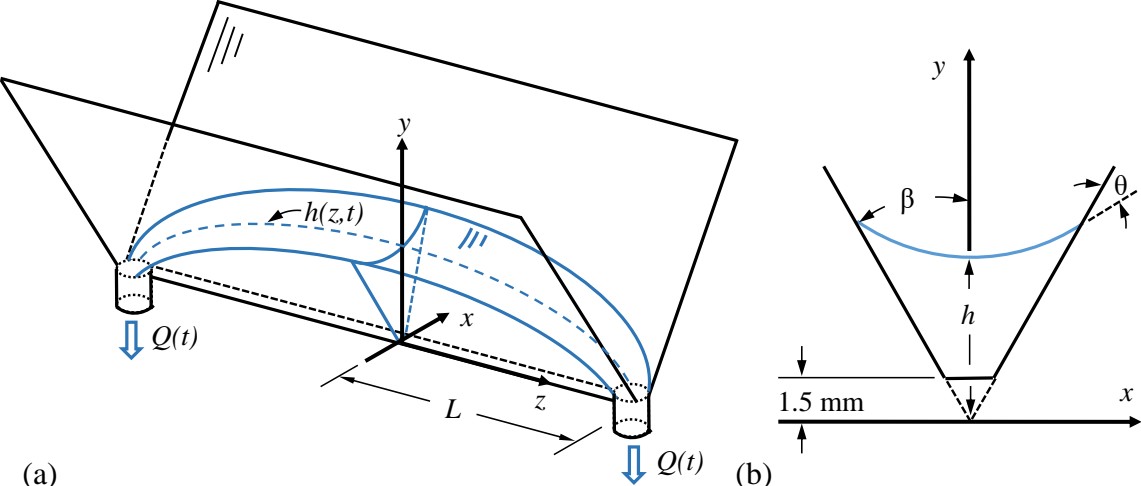

**Figure 2.** Schematic of double-drain flow analytical and computational domain. Drain ports are symmetric at $z = \pm L$, where volumetric sink flow rates $Q(t)$ are specified: (**a**) perspective view and (**b**) cross-sectional view, dashed end cropped from computational domain.

## 2. Background

The capillary driven flows of this investigation are created by underpressure gradients in the liquid caused by positive streamwise curvature gradients in the liquid free surface. Such gradients

result from forced flows or in many cases spontaneous wicking flows determined by initial conditions. All such curvature gradients arise as the liquid seeks to establish contact angle boundary conditions at the liquid–solid–gas line of contact (i.e., the wall of the container). The 'contact line' is thus a critical boundary condition. Unfortunately, for the contact line to move, it must appear to violate the no slip condition. This quandary has received significant attention over many decades from theoretical [8,9], experimental [10,11], and numerical [12,13] perspectives. The concern is exaggerated in low-g environments where uncertainties in the essentially nanoscale region of the contact line can influence literally tons of liquids at the $O(1m)$ scale. Practical numerical models of such flows treat shortcomings in our physical understanding of the moving contact line with a variety of simplifications in an attempt to discretize competing mathematical models of the phenomena. We highlight a limited selection of recent research that applies most directly to our numerical method.

For example, while high Capillary number oscillatory flows require dynamic contact angle models [3,14], recent studies in similar capillary regimes to those studied herein report encouraging, if not surprising, constant contact angle simulation results. Gurumurthy et al. [15] investigate the spontaneous rise of a liquid in an array of open rectangular channels, where OpenFOAM simulations agree well with power-law theoretical predictions. Malekzadeh and Roohi [16] study flow behavior and droplet formation in T-junction micro-channels, where OpenFOAM simulations agree well with micro-channel experiments. Yong-Qiang et al. [17] simulate capillary rise in fan-shaped interior corners, such that each wall of the corner has a specified constant contact angle not necessarily that of its neighbor, where simulations carried out in FLOW-3D, compare well to drop tower experiments. Arias and Montlaur [18] analyze bubble generation in a capillary T-junction geometry, and report that, while contact angle is a sensitive parameter for low Capillary number $O(10^{-3})$, for large Reynolds number $O(10^2 - 10^3)$ flows, less than 4% error is maintained for all measured quantities between ANSYS VOF simulations and micro-channel experiments. Klatte et al. [19] simulate parabolic flight and drop tower capillary drain-fill wedge geometries in microgravity using a pressure potential field within the static equilibrium Surface Evolver algorithm [20], reporting agreement within 2% for all measured quantities. Such simulation agreement with experiments and theory for a variety of capillary-dominated flow regimes gives confidence to the simple use of the static contact angle model. As such, the simulations reported herein assume a constant contact angle.

Regarding our numerical approach, the VOF method implemented in OpenFOAM's interFoam flow solver produces an interface stretching over a few computational cells [21]. This non-sharp volume fraction can lead to curvature errors, inducing non-physical spurious currents [22]. These stem from the inability of the surface tension algorithm to evaluate a constant interface curvature, thereby generating non-physical capillary waves [23]. While level set methods can be shown to run faster and more accurately calculate curvature, mass-conservative schemes remain challenging [24]. A recent comprehensive review of spurious currents in VOF and level set methods was conducted by Popinet [25]. Soh et al. [26] study droplet formation in T-junction micro-channels, where smoothing operations applied to OpenFOAM's VOF model minimize spurious velocities, improving simulation-experimental agreement. Sontti et al. [27] employ a coupled level set and VOF method, reporting reduced spurious currents compared to the sole VOF counterpart, thereby in better agreement with the micro-channel experiments. Guo et al. [28] replace the ANSYS continuum surface force model with a modified height function method, resulting in increased accuracy for micro-channel annular flows. While the state of the art is ripe with techniques to minimize spurious currents, prior to simulating transient drains, we first analyze a static liquid supported in a wedge geometry in a zero-gravity environment where we find negligible spurious velocities, adding confidence to the reported simulations.

## 3. Experiments and Data Reduction

The Capillary Flow Experiments (CFE) conducted aboard the ISS were a series of handheld, large length scale (<20 cm) experiments. CFE was pursued to provide data for analytical and numerical model development for capillary flow phenomena relating to moving contact line boundary conditions, critical geometric wetting, and interior corner flows. The latter were pursued via 9 Interior Corner Flow (ICF) test vessels (ICF-1, ICF-2, ..., ICF-9), each representing a geometry of practical concern. A video archive for the ICF test suites is publicly available through login at the mainpage https://psi.nasa.gov/, mission narrative explained at https://www.nasa.gov/sites/default/files/atoms/files/psi_researchers_guide-tagged.pdf. The double-drain tests analyzed herein were conducted in late 2016 and early 2017. Figures 3 and 4 provide an annotated image, solid model, and wire model for the ICF-1 and ICF-8 test vessels respectively. For each test, a given ICF vessel is placed on an ISS workbench, back-lit by a diffuse light screen via cabin lighting, and filmed via an HD Canon XF305 video camcorder fabricated in Tokyo, Japan. During the approximately 3 h of manual crew interaction with each ICF vessel on the ISS, a suite of capillary drain tests were performed. The astronauts drain liquid from the Tapered Section into the Reservoir by turning the Piston Dial counterclockwise. While draining, capillarity wicks fluid into the Interior Corner, the central geometric drain element spanning the vessel length. The drain tests reported herein were performed with Control Valve 2 fully open while Control Valve 1 was adjusted to approximately balance draining resistances in both valves. In this manner, astronauts evenly drained both sides of the Interior Corner, minimizing meniscus height at the two vessel outlets without ingesting gas and emulating a double-drain similar to Figure 1 (right), shown schematically in Figure 2.

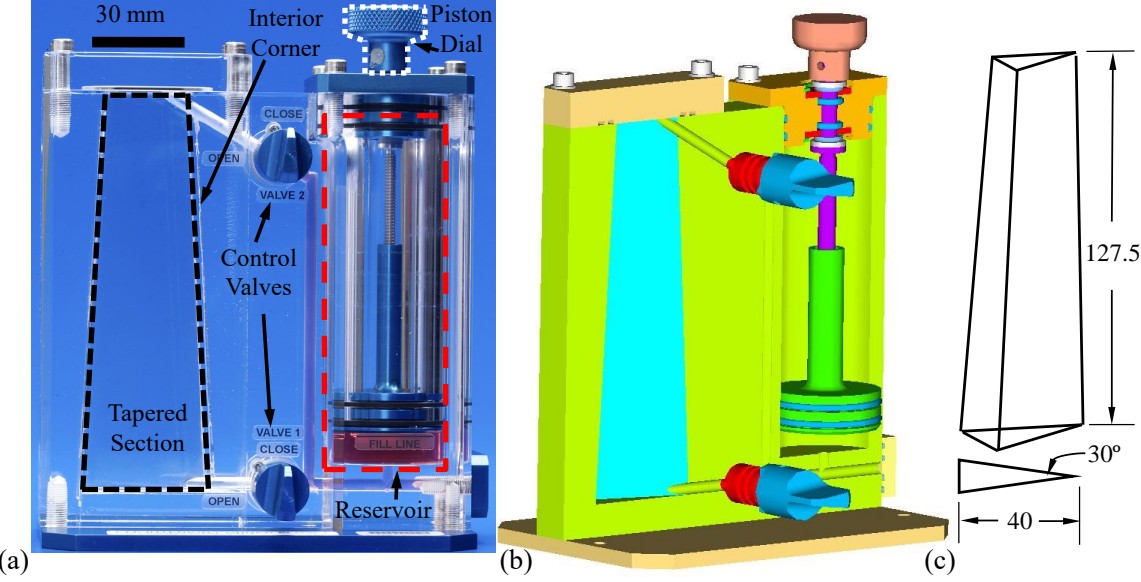

**Figure 3.** Annotated ICF-1 test Vessel: (**a**) apparatus, (**b**) solid model, and (**c**) wire model with dimensions in mm, where large (**bottom**) to small (**top**) cross-sectional isosceles triangles are congruent through a 20:13 ratio.

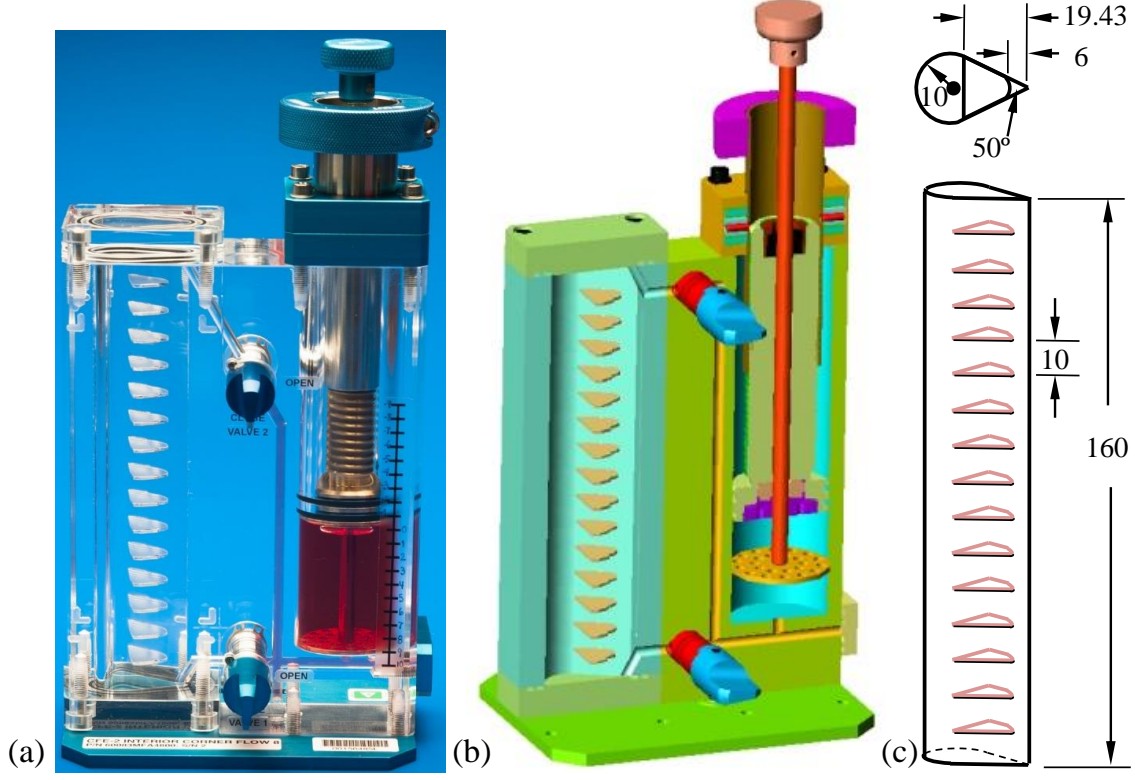

**Figure 4.** Annotated ICF-8 test Vessel: (**a**) apparatus, (**b**) solid model, and (**c**) wire model with dimensions in mm.

Digitized time-dependent interface profiles are readily collected from the ICF data with which comparisons to analytical and numerical predictions are straightforward. We pursue this tack herein following a brief review of a simple analysis applied to the subject double-drained interior corner. We eventually increase the geometric complexity of the corner such that the simple analytical model below no longer applies, but the experimental and numerical comparisons remain immediate.

Further details of the experimental data reduction process are briefly discussed here. The primary objective is to digitize the experimental fluid interfaces $h_e$ to determine and compute meniscus profiles and instantaneous drain rates. In-house interfacial tracking algorithms were developed to extract the interface position from the images. The experimental volume $V_e$ remaining in the Tapered Section was then computed via a meniscus integration over $h_e$ using

$$V_e = F_A \int_{-L}^{L} h_e^2 \, dz, \tag{1}$$

where $F_A$ is a geometric constant reported in Table 1, $L$ and $z$ shown in Figure 2. There are two ways to establish the experimental drain rate $Q_e$: (1) time rate of change of $V_e$, and (2) transient piston position during the drain process, Figure 3a. The latter is not sufficiently accurate to determine $V_e$: the image pixel resolution combined with the large piston area can yield measurement errors several times larger than the measured quantity, Figure 5a. Figure 5b presents a sample volume integration method for $Q_e$. The volumetric drain rates at experimental frame times were smoothed via trapezoidal interpolation, maintaining the overall volume-time integral average, but smoothing the drain rates. Figure 5b plots raw and smoothed values for $Q_e$. In this manner, draining appears continuous despite the nearly peristaltic

method applied by the astronauts (i.e., nearly continuous counterclockwise hand turn of the Piston Dial). Total drain time is 70 s for ICF-1 and 69 s for ICF-8. We note $h_e$ was not tracked near the container edge due to light reflection impeding the interfacial tracking algorithm.

**Table 1.** ICF-1 fluid properties, scales, and constraints. Dual values listed as (experiment, simulation) when different from each other.

| Property | Units | ICF-1 | ICF-8 |
|---|---|---|---|
| Density, $\rho$ | kg m$^{-3}$ | 950 | 910 |
| Viscosity, $\mu$ | kg m$^{-1}$ s$^{-1}$ | 0.0190 | 0.00455 |
| Surface tension, $\sigma$ | N m$^{-1}$ | 0.0206 | 0.0197 |
| Contact angle, $\theta$ | deg | $0°$ | $0°$ |
| **Scales** | **Units** | **ICF-1** | **ICF-8** |
| Half angle, $\beta$ | deg | $15°$ | $25°$ |
| Flow length, $L$ | mm | 63.5 | 80 |
| Initial half volume, $V_i$ | mm$^3$ | 1468, 1484 | 10,096, 10,549 |
| Height, $H = \sqrt{V_i/F_A L}$ | mm | 8.8, 8.9 | 15.3, 15.6 |
| Velocity, $W = \sigma \epsilon F_i \sin^2 \beta / \mu f$ | mm s$^{-1}$ | 4.5 | 31.5, 32.2 |
| Flow rate, $Q \sim W F_A H^2$ | mm$^3$ s$^{-1}$ | 105.0, 106.2 | 3970, 4241 |
| Time, $t \sim L/W$ | s | 14.0 | 2.54, 2.49 |
| **Lubrication Assumptions** | **Constraint** | **ICF-1** | **ICF-8** |
| Slender geometry, $\epsilon = H/L$ | $\epsilon^2 \ll 1$ | 0.0194, 0.0196 | 0.0364, 0.0381 |
| Low streamwise curvature | $\epsilon^2 f \ll 1$ | 0.0068 | 0.0267, 0.0279 |
| Capillary dominance | $Bo \ll 1$ | $\sim 10^{-4}, 0$ | $\sim 10^{-4}, 0$ |
| Low intertia | $\epsilon^2 Su F_i^2 \sin^4 \beta / f \ll 1$ | 0.0029, 0.0030 | 0.5111, 0.5340 |
| Concus-Finn wetting | $\theta < 90° - 2\beta$ | satisfied | satisfied |

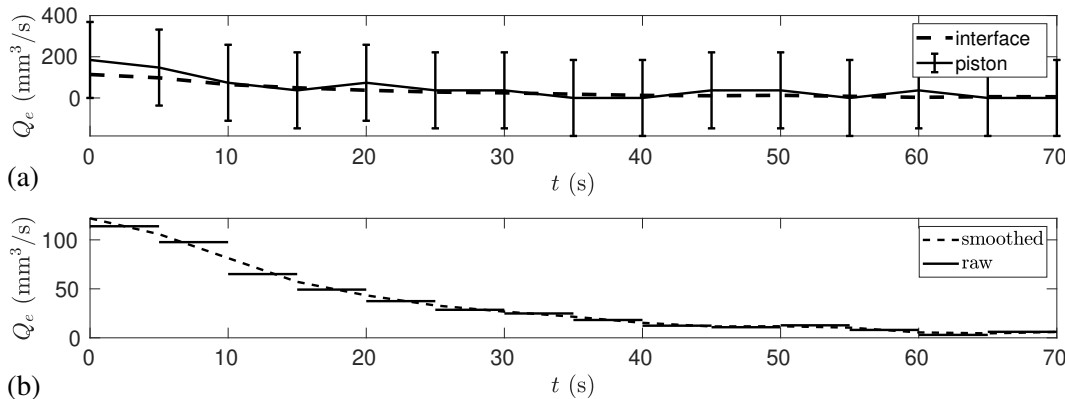

(a)

(b)

**Figure 5.** ICF-1 $Q_e$ (**a**) measurement error for piston (dashed, box-whisker) and interfacial (dashed) tracking, where whisker length and dashed line thickness indicate measurement error for each respectively and (**b**) trapezoidal-integrated smooth (dashed) alongside raw (discontinuous black, solid).

## 4. Numerical Model Review

The ICF test cell sections were stenciled in Fusion 360, a CAD Autodesk tool. The stencil was meshed in snappyHexMesh, refining hexagonal cell layers near the surface geometry. The initial cell count was 42,571 cells for ICF-1 and 121,860 cells for ICF-8, with a one layer dynamic mesh refinement imposed at the interface (8 cell sub-refinement per interfacial cell). The simulation run time was approximately 20 and

80 h for ICF-1 and ICF-8, respectively on an AMD EPYC 7281 Hexadeca-core (16 core) processor running parallel on all 16 cores. Initial conditions were specified via swak4foam, detailed below. Transverse and axial symmetry boundary conditions were implemented in the $z = 0$ and $x = 0$ planes, respectively (Figure 2), reducing the computational domain to 1/4 the Tapered Section. The vessel walls impose no slip and a constant contact angle. Since lubrication approximations are satisfied (Table 1), and by choice of scales that absorb many dynamic contact angle geometric effects [29], a constant contact angle model is qualified [30,31]. The exit port imposes a uniform velocity that linearly interpolates specified velocities temporally. To conserve mass, an atmospheric boundary condition is imposed on a rectangular portion of the vessel roof. The sharp lower interior corner $y \in [0, 1.5]$ mm of the Tapered Section was removed for efficient meshing and run times (Figure 2b). For ICF-1, the upper portion ($y > 18$ mm) of the section was removed for computational efficiency. These assumptions had negligible effects on the numerical results.

The interFoam solver is chosen for this laminar, incompressible, two-phase fluid flow. A VOF technique at the free surface is prescribed with scalar indicator function $\alpha \in [0, 1]$ such that $\alpha = 0$ implies the computational cell is gas and $\alpha = 1$ implies the computational cell is liquid. All cells average fluid properties based on the volume fraction fraction of liquid, using density $\rho$ as an example:

$$\rho = \alpha \rho_l + (1 - \alpha)\rho_g \tag{2}$$

where subscripts $l$ and $g$ denote liquid and gas, respectively. Incompressiblity implies $\alpha$ satisfies the advection equation:

$$\partial_t \alpha + \nabla \cdot (\alpha \boldsymbol{u}) = 0 \tag{3}$$

where $\boldsymbol{u} = \langle u_x, u_y, u_z \rangle$ is the fluid velocity. Continuity and momentum equations governing the flow are

$$\nabla \cdot \boldsymbol{u} = 0, \tag{4}$$

$$\partial_t (\rho \boldsymbol{u}) + \nabla \cdot (\rho \boldsymbol{uu}) = -\nabla P + \nabla \cdot \left( \mu \left( \nabla \boldsymbol{u} + \nabla \boldsymbol{u}^T \right) \right) + \boldsymbol{F}_b, \tag{5}$$

for pressure $P$ and dynamic viscosity $\mu$. Force $\boldsymbol{F}_b = \sigma \kappa \nabla \alpha$ is the surface tension body force vector modeled by the continuum surface force method of Brackbill et al. [32], where $\sigma$ is the liquid–gas surface tension and $\kappa = -\nabla \cdot (\nabla \alpha / |\nabla \alpha|)$ the liquid–gas interfacial curvature. The aforementioned equations of motion are well-posed once an initial condition is specified. ICF-1 imposed a constant, initial height $h_s = 9.3$ mm, determined to match the initial experimental volume. ICF-8 imposed a constant initial height $h_s = 17.2$ mm, where draining was suspended for the next 3 s to allow the liquid to fill the vanes and establish a static equilibrium where experimental and simulation heights agree, at which point draining begun. Time $t = 0$ corresponds to the time draining ensues. The PIMPLE algorithm is applied for pressure–velocity coupling with increased stability (nCorrectors set to 3). Time integration is performed via first order Euler scheme, allowing a dynamic time step bounded by maximum Courant number 0.2 [3,14–16]. A first order Gauss linear scheme discretizes the gradient terms. Second order Gauss linear, upwind, and vanLeer schemes discretize the divergence terms.

A mesh-independence study was conducted to demonstrate spacial convergence and justify cell refinement. Figure 6 presents time averaged simulation peak center-line interfacial height $h_s$ and time averaged volumetric flow rate $Q_s$ (reference Figure 2a) as percent errors against the base case 42,571 cell count ICF-1 vessel. Both quantities plotted are functions of the percent of the initial 42,571 cell count. We see that increasing the cell count by 10% has less than 1% change in reported values. Then, we conclude the reported results are mesh-independent.

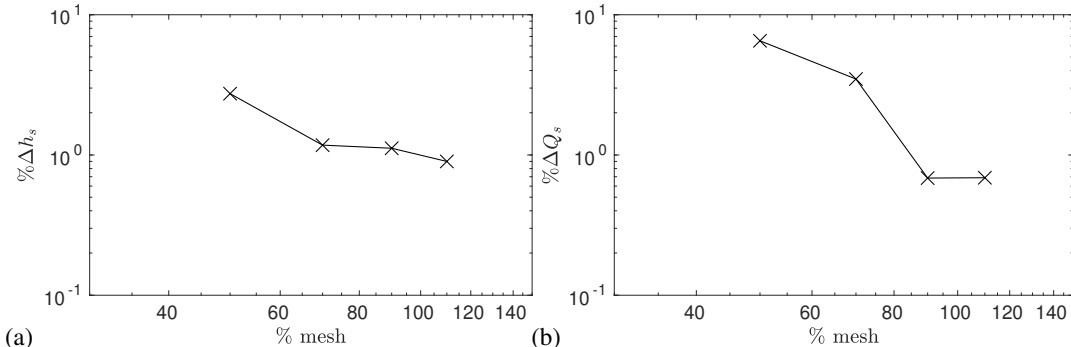

(a)      (b)

**Figure 6.** Convergence plots, where (**a**) percent height error $\%\Delta h_s$ and (**b**) percent volumetric flow rate error $\%\Delta Q_s$ are compared against the ICF-1 analytic study as functions of the reported percent mesh count. Values time-averaged for the first 15 s of draining.

Saufi et al. [33] study spurious currents in the interFoam solver, placing a 1 mm diameter water droplet in a medium. After 0.2 s, improper curvature calculations resulted in catastrophic droplet deformation. For the ICF-1 vessel, we conduct a similar study: draining is switched off and a slab of liquid is placed in the wedge. Sufficient time is given for the liquid to wick through the corners until static equilibrium is maintained. We then compute the Capillary number $\mu V/\sigma$ at each time step, where here $V = \|\boldsymbol{U}\|_\infty$ over the entire liquid domain. For the next 20 s, we report all Capillary numbers less than 0.02. Figure 7 plots interfacial velocity vectors at their corresponding locations and the underlying liquid for several times. Clearly, the interface is stable, owing to the wedge support. Thus, while infectious to numerous flow problems, we report small spurious currents that are unlikely to significantly cloud the simulation results.

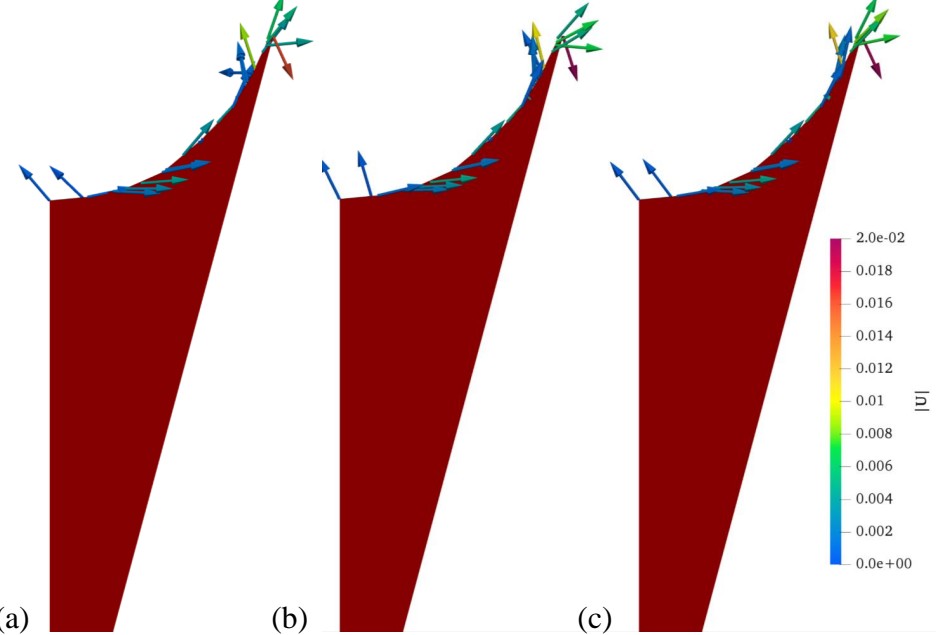

(a)      (b)      (c)

**Figure 7.** Part of ICF-1 cross-sectional slice with liquid (red) in equilibrium, normalized velocity $|\boldsymbol{U}|$ interfacial vectors with colorbar magnitude (not pertaining to solid red liquid) at (**a**) 0 s, (**b**) 10 s, and (**c**) 20 s.

## 5. ICF Experiments and Simulations

### 5.1. Hypothesis

This study attempts to validate the working hypothesis: *OpenFOAM's non-modified interFoam solver accurately predicts large length scale (centimetric) capillary drainage in wedge geometries.* 'Accurate' is assessed via analyzing three fundamental parameters described in detail below: peak interfacial height $h$, volumetric flow rate $Q$, and meniscus evolution $h(z, t)$. We first benchmark the simulations against a simple analytical corner drain model, Section 5.2. Closed-form expressions (6)–(10) are readily compared to the simulations. We then simulate two flight experiments: smooth wedge walls ICF-1 Section 5.3, and the wedged-vane network of ICF-8, Section 5.4. An analytical model is also compared to both where appropriate. We assess the aforementioned quantities and compare simulations to experiments.

### 5.2. Simplified Analysis

A single symmetric double-drained interior corner is sketched in Figure 2. Symmetric flow is assumed with liquid volumetric flow rate $Q(t)$ specified at each drain port $z = \pm L$, where the meniscus height $h(\pm L, t) = 0$ is specified. If the liquid wets the corner such that the Concus–Finn corner wetting condition is satisfied [6] (Table 1), the liquid spontaneously wets into and along the interior corner. As draining at $z = \pm L$ ensues, the capillary pressure becomes increasingly negative via $P \sim -1/h(z, t)$, and liquid migrates by capillarity toward the drain ports where depth $h$ is shallowest. Following [7], all variables are non-dimensionalized according to Table 1 values and presented as dimensionless unless otherwise specified. We invoke subscripts $e, s, a$ to denote experimental, simulation, and analytic analysis values, respectively. For a long narrow channel satisfying the constraints of Table 1, an asymptotic lubrication prediction for $h_a(z, t)$ is

$$h_a(z, t) = \frac{F(z)}{t_i + t/t_i}, \tag{6}$$

$$F(z) = \left( a_0(1-z) - \frac{27}{20} a_0^{2/3} (1-z)^{8/3} + \frac{243}{650} a_0^{1/3} (1-z)^{13/3} \right)^{1/3}, \tag{7}$$

where $t_i = 2.2059$ is a constant determined by the dimensionless initial half-domain volume and $a_0 = 729(10 + 7\sqrt{2})/500$ is a dimensionless constant determined by the meniscus slope at the drain port. Solutions for respective interface peak height $h_a(z = 0, t)$, liquid half-domain volume $V_a(t)$, and liquid half-domain volumetric flow rate $Q_a(t)$ are given by

$$h_a(z = 0, t) = \frac{1.178}{1 + t/t_i}, \tag{8}$$

$$V_a(t) = \frac{1}{(1 + t/t_i)^2}, \tag{9}$$

$$Q_a(t) = \frac{0.907}{(1 + t/t_i)^3}. \tag{10}$$

The model above assumes idealized exit conditions and geometry. Such a flow may also be simulated via OpenFoam using the geometric assumptions identified in Figure 3 for the ICF-1 vessel geometry. Excellent agreement is found for the simplified analysis and OpenFoam simulations for the first 30 s of drain process: $|h_s(z = 0) - h_a(z = 0)| < 6.5\%$ and $|Q_s - Q_a| < 4.5\%$ (Figure 8a,b). During the simulation, at approximately $t = 30$ s the drain port ingests gas, significantly retarding $Q_s$. The time at ingestion is shown in Figure 8b (dotted line). Figure 8c plots (7) against $h_s \cdot (t_i + t/t_i)$ at half-second intervals

for the first 30 s of draining. The collapse of simulation profiles demonstrates the numerically verified self-similarity of the flow as suggested by the simple theory (6).

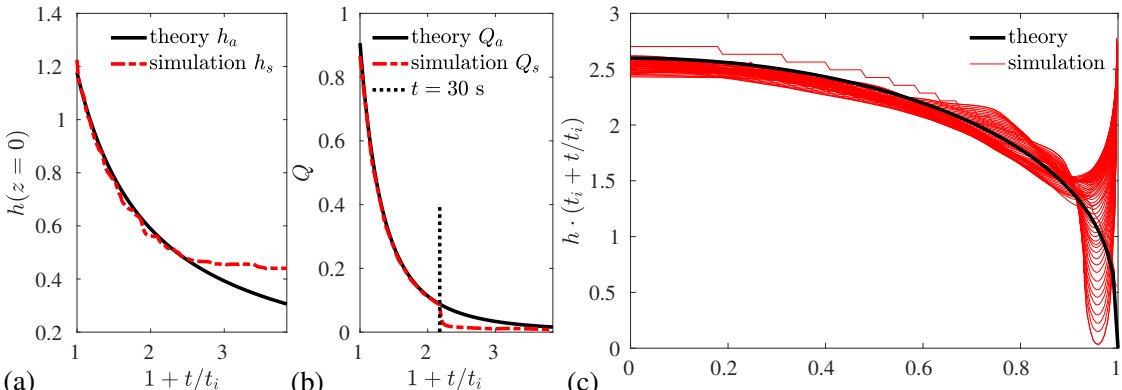

**Figure 8.** (**a**) $h_s$ (dashed) and $h_a$ (8) (solid) plotted against time, (**b**) $Q_s$ (dashed) and $Q_a$ (10) (solid) plotted against time, with ingestion time $t = 30$ s (dotted). (**c**) $h \cdot (t_i + t/t_i)$ (red) profiles plotted against lubrication theory (7) (black) at half second intervals for the first 30 s of draining. Simulation initial and boundary condition match lubrication prediction.

### 5.3. Comparison of Numerics and Analysis with Experiments for ICF-1

Figure 8 presents simulation results with the simple theoretical predictions (6)–(10). The analytical model assumes a point sink for the drain port, which excludes the possibility of gas ingestion. For drain ports with finite radii (ICF-1), a zero meniscus height specified at the drain port implies that gas ingestion is imminent. To circumvent immediate gas ingestion, the simulation initial condition incorporates a thin film $\epsilon_0$, $h_s(z, t = 0) = F(z)/t_i + \epsilon_0$ in (7), where $\epsilon_0 = 0.5$ mm. A uniform velocity boundary condition is prescribed at the drain port $U_{vel} = V_a'(t)L/(V_i W \pi r^2)$, where $r = 2$ mm radius drain port, scales defined in Table 1. This velocity condition is consistent with the analytical model prediction, which enforces $h_s(z = L, t) = 0$. The experimental results of ICF-1 are also presented, the computational domain shown in Figure 9.

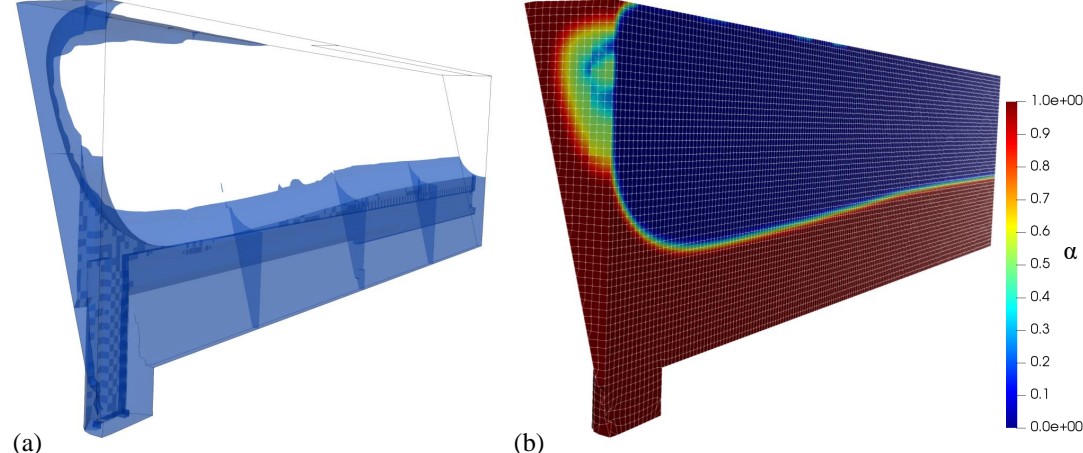

**Figure 9.** ICF-1 computational domain showing (**a**) liquid volume as phase fraction $\alpha < 0.6$ and (**b**) phase fraction $\alpha$ interpolated throughout each computational cell (outlined in white).

Figure 10 shows a still image of ICF-1 experimental draining alongside a 3D cross-sectional simulation image for qualitative comparison. Figure 11a,b presents simulation, experimental, and simple analytical results. Lack of gas ingestion is observed for the full elapsed drain time, where excellent simulation-experimental volumetric flow rate agreement is seen: $|Q_s - Q_e| < 4\%$, Figure 11b. Meniscus height $h_s$ agrees well with $h_e$ but coincidence is not expected: $h_s(t = 0)$ was determined to satisfy $V_a(t = 0) = V_e(t = 0)$. Additionally, the simulations are drained unevenly about $z = 0$; as such, for early times $h_e(z = -L) < h_s(z = -L)$ and $h_e(z = L) > h_s(z = L)$. Despite these discrepancies, $|h_s(z = 0) - h_e(z = 0)| < 10\%$. At $z = \pm L$ the experimental flows were observed to wick up the edge toward the upper surface of the vessel, shown in Figures 10a and 11c. This expected migration owes to the PDMS liquid perfectly wetting ($0°$ contact angle) the vessel walls. The analytical model predictions are shown to increasingly under-predict $h_e$ at late times, as evidenced in Figure 11a. This expected result owes to the $h_a(z = \pm L) = 0$ boundary condition.

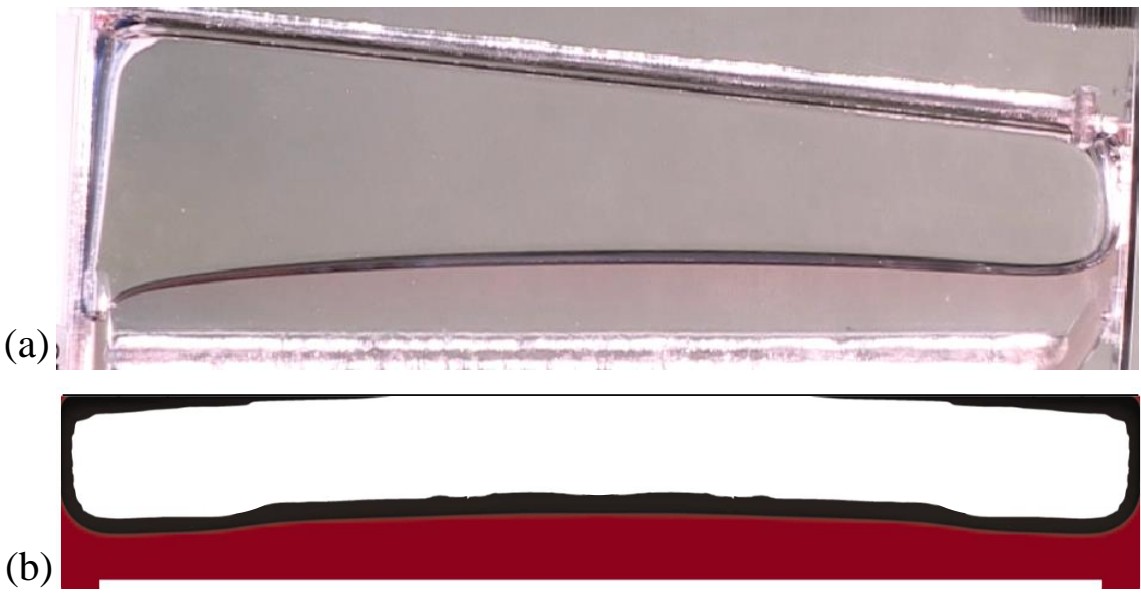

**Figure 10.** Profile view of the ICF-1 Tapered Section filled with liquid (red) at $t = 10$ s (**a**) during experiment and (**b**) simulation, where gray implies interfacial volume fraction $\alpha = 0.5$.

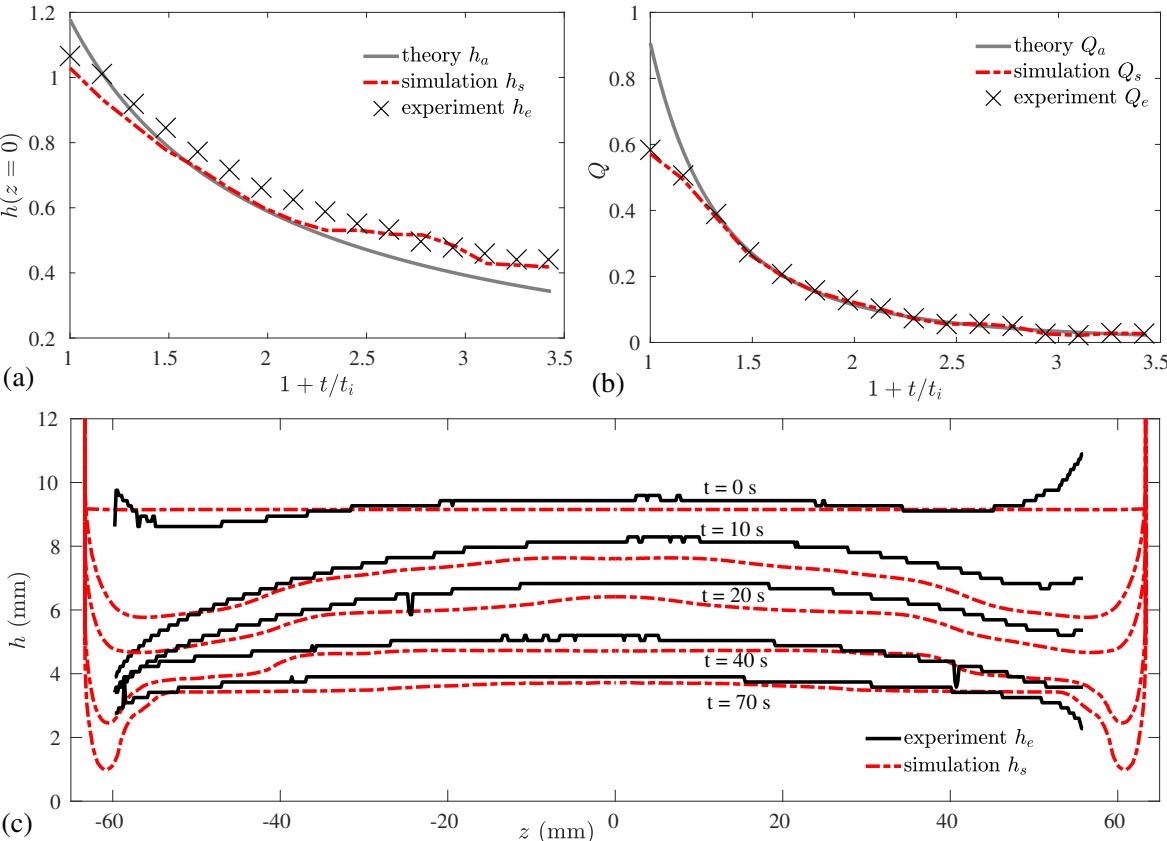

**Figure 11.** (**a**) $h_s$ (dashed), $h_a$ (8) (solid), and $h_e$ (ticks) plotted against time, (**b**) $Q_s$ (dashed), $Q_a$ (10) (solid), and $Q_e$ (ticks) plotted against time; (**c**) dimensional $h_s$ (dashed) plotted against $h_e$ (solid) at specified times.

### 5.4. Comparison of Numerics with Experiments for ICF-8

In a similar manner to the ICF-1 vessel, an image of ICF-8 was shown in Figure 4. In this container, a snow-cone cross-sectioned interior corner test cell is further partitioned by vane segments along the $z$-axis as detailed in Figure 4c. This vessel was designed to study bubble separation characteristics in such geometries, but the double-drain flow could be established and was demonstrated on the ISS. The corner flow again provides symmetric draining of the container with the vane segments adding geometric complexity by introducing a second stream-wise curvature component to the free surface. A sample draining event is shown in Figure 12. The verified OpenFoam software was re-programmed for the ICF-8 geometry. Simulation results are compared to the ICF-8 experiment data in Figure 13. The initial condition for the numerical computations of $h_s = 18$ mm was given 3 s to relax to equilibrium, as $\theta = 0°$ induces wicking along the edges and notably within the vaned segments of the container. The initial height $h_s = h_e$ was also selected resulting in the mismatched initial volumes, Table 1, which is attributable to image analysis limitations detecting the meniscus edge everywhere in the container (i.e., side walls and upper corners). To address this issue, we non-dimensionalize ICF-8 computational results with $V_e$. The following analysis reports draining at $t = 0$ s. The total duration of the drain time of this test is 69 s.

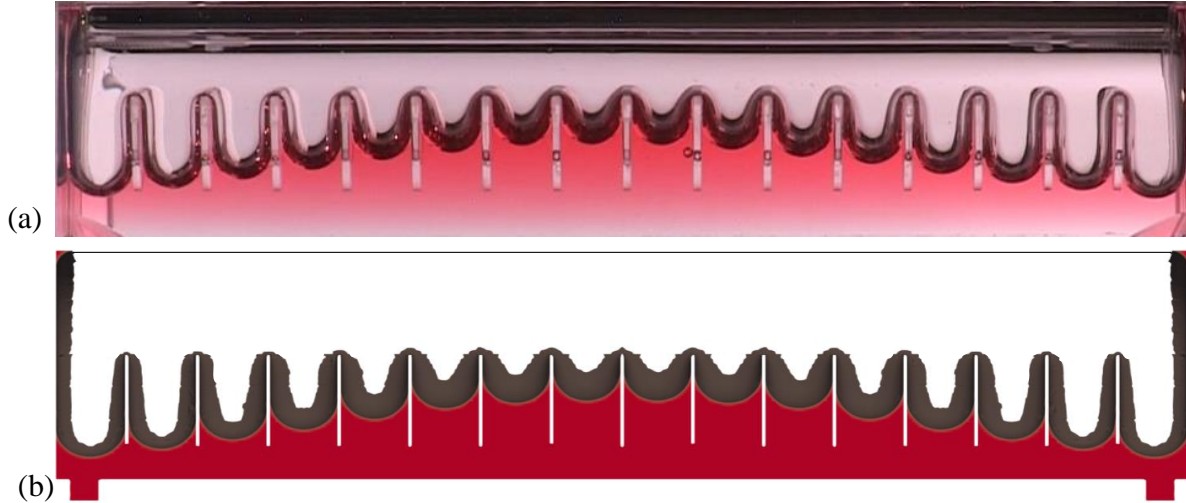

**Figure 12.** 3D side perspective of the ICF-8 Tapered Section filled with liquid (red) at $t = 25$ s (**a**) during experiment and (**b**) simulation, where gray implies interfacial volume fraction $\alpha = 0.5$.

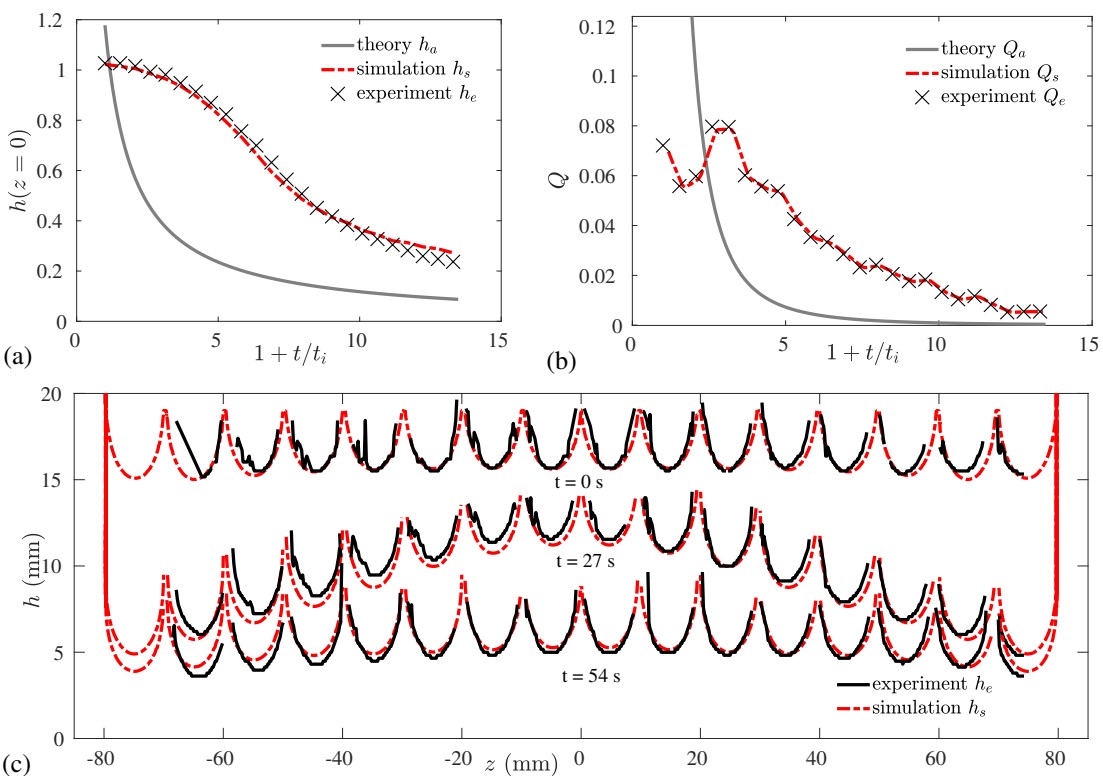

**Figure 13.** (**a**) $h_s$ (dashed), $h_a$ (8) (solid), and $h_e$ (ticks) and (**b**) $Q_s$ (dashed), $Q_a$ (10) (solid), and $Q_e$ (ticks) with time. (**c**) Dimensional $h_s$ (dashed) against $h_e$ (solid) at specified times.

Figure 13a,b presents comparisons of the simulations and experimental results. The absence of gas ingestion is observed for the full elapsed drain time, where excellent simulation-experimental volumetric flow rate agreement is seen: $|Q_s - Q_e| < 4.5\%$, Figure 13b. Meniscus height error is reported $|h_s - h_e| < 16\%$ for the full 69 s drain, where we note $|h_s - h_e| < 6\%$ for the first 60 s. The analytic model for a

simple interior corner is also presented for token comparison despite the significant violations of model assumptions attributed to the vane-segment geometry.

## 6. Discussion

The Bond number $Bo \equiv \rho g H^2 / \sigma$ and Suratman number $Su \equiv \rho \sigma H / \mu^2$ determine the relative strength of hydrostatic pressure to capillary pressure and capillary inertia to viscous resistance, respectively. Values for these critical dimensionless groups are listed in Table 1. The dimensionless parameters for viscous resistance $F_i = 0.1560$, section area $F_A = 0.2955$, and surface curvature $f = 0.3492$ depend on system geometery only through corner half-angle $\beta$ and liquid contact angle $\theta$. Further details are provided in [7].

In reference to Figure 11b, at small amounts of time, ICF-1 $Q_a$ nearly doubles $Q_e$. Defining time $t = 0$ s as the moment the astronaut began draining the test cell, the video shows several seconds before approximate zero height is realized: the astronaut drained the container much slower than the simple analytical model prediction assumes. For large amounts of time, self-similarity is achieved, as $Q_a$ closely predicts $Q_e$. Though disagreement is exaggerated for the analytical model at early times, the OpenFOAM simulations accurately capture $Q_e$ for all time. In late stage draining, $h_a$ under-predicts $h_e$.

The analytic model works well to predict the draining flow in ICF-1, but predictable errors are observed when the model is inappropriately extended to the draining flow of ICF-8. In general, the simple analytical model assumes negligible inertia. Table 1 lists the order of magnitude expectations of inertia for both ICF-1 and ICF-8 flows, where it is observed inertia is non-negligible for ICF-8. Figure 13b corroborates this, as $Q_e \gg Q_a$. The simulations account well for finite inertia, accurately predicting $h_e$ and $Q_e$ with near coincident menisci observed over a range of drain times, Figure 13c. While it is clear that the linear analytical model does well for simple interior corners, the OpenFOAM simulations are extendable to channel networks of greater geometric complexity and higher flow inertia.

## 7. Conclusions

OpenFOAM capillary drain simulations were validated against spacecraft flight experiments conducted as part of the Capillary Flow Experiments on the International Space Station. The flow scenario was that of the symmetric late stage draining in interior corner networks of varying geometric complexity including a straight interior corner (ICF-1) and one segmented by a network of vane elements (ICF-8). A simple analytical model for the flow is also used to benchmark the numerical method, where better than 6.5% agreement is found for all measured quantities prior to gas ingestion, occurring approximately 30 s into draining. Experimental meniscus heights $h_e$ and liquid volumetric flow rates $Q_e$ agree with the OpenFOAM simulations to within 10% and 5% respectively for the first 60 s of draining. The simple analytical model is found to under-predict the meniscus height and volumetric flow rate, and, while providing reliable height and drain rate predictions for simplified geometries, serves better as a lower-bound for complex vane networks. To this end, OpenFoam's interFoam solver is found to serve as an excellent tool for analytical model verification as well as quantitative drain rate assessment particularly for flows of increased geometric complexity and inertia.

**Author Contributions:** Conceptualization, J.M.; Data curation, J.M.; Formal analysis, J.M.; Investigation, J.M.; Supervision, P.S.; Writing—original draft, J.M.; Writing—review & editing, M.W. All authors have read and agreed to the published version of the manuscript.

**Funding:** This research was funded primarily by NASA under NNH17ZTT001N-17PSI D. M.W. is supported in part through NASA Cooperative Agreements 80NSSC18K0161 and 80NSSC18K0436.

**Acknowledgments:** Special thanks to JAXA astronaut K. Wakata for his patience and creativity during flight experiments. This paper is dedicated to the late Paul Steen, 6/22/1952–9/4/2020.

**Conflicts of Interest:** The authors declare no conflict of interest.

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
