# Peer review of "OpenFOAM Simulations of Late Stage Container Draining in Microgravity"

_fluids, doi:10.3390/fluids5040207_

Round 1

Reviewer 1 Report

This paper compares the numerical simulations with OpenFOAM of a system for symmetric draining of capillary liquids in simple interior corners with experimental results obtained in the ISS and a simple analytical model.
Results show a good agreement between the OpenFOAM simulations and the experiment, validating possible use of simulations for complex geometries that can not be solved with the simple analytical model.

The paper is clear and well written, but some parts need to be much more detailed, mainly: the literature review and the hypothesis of the OpenFOAM simulations.

Introduction: The introduction needs to be completed with a proper state of the art of the kind of CFD simulations presented here. In particular, examples of zero(micro)-g numerical simulations made with OpenFOAM could be included, as well as VOF simulations made with OpenFOAM.
Examples of VOF (and/or) zero(micro)-g numerical simulations made with other CFD software should also be cited. In particular, papers tackling the wall contact angle problem in these conditions are of interest in this literature review.
Some examples:
Malekzadeh, S., Roohi, E.: Investigation of different droplet formation
regimes in a T-junction microchannel using the VOF technique
in openFOAM. Microgravity Sci. Technol. 27, 231–243 (2015).
https://doi.org/10.1007/s12217-015-9440-2

Yong-Qiang, L., Wen-Hui, C., Ling, L.: Numerical Simulation
of Capillary Flow in Fan-Shaped Asymmetric Interior Corner Under Microgravity. Microgravity
Sci. Technol. 29, 6579 (2017)

Arias, S., Montlaur, A.: Influence of contact angle boundary condition
on CFD simulation of T-junction. Microgravity Sci. Technol.
30(4), 435–443 (2018). https://doi.org/10.1007/s12217-018-96
05-x

Then, I understand that the main focus of the paper is to present the OpenFOAM results and compare them with the experiment (+ analytical model). Nevertheless, the experiment is extremely interesting and though it is described here, if it has been previously published (whether in a NASA report or in a Journal) I think it should also be included (cited) in the introduction.

p.2 line 43: I would advise to properly cite the report found here: https://www.nasa.gov/sites/default/files/atoms/files/psi_researchers_guide-tagged.pdf
which will be more permanent that the picture in the PSI main page. Or, as previously commented, feel free to cite other NASA report referring to this experiment (I am just suggesting one I found)
I would also suggest adding the date (and/or mission name) of when the experiment in the ISS took place.

p.6: please provide more information about the numerical simulation setup, namely:
1. detail the initial conditions considered (from swak4foam): they come in 3.2, they would probably be better before the results part.
2. contact angle boundary conditions are key in this type of problem. Please comment your choice and, as previously commented, refer to some literature works on this specific topic.
3. Apart from the number of elements, it would be good to have some information concerning the mesh size. Have you done a convergence study for the mesh size? It would be interesting to comment it, to compare the level of error that you obtain with the expected precision of your simulation (due to the mesh size).
4. Have you checked that a dynamic time step gives you better time convergence than a fixed one?

Overall I would suggest adding a dedicated section (sub section) to OpenFOAM hypothesis, summary of main equations, etc. It would clarify a lot this part.

p.7 l.138: Can you add a picture similar to Figure 9a, for the experiment ICF-1? It would be good to show how the flow wicks up along the edge. It can't really be appreciated in Figure 8c.
And you can further comment about how the 0º contact angle perfectly reproduces (as expected) this wick-up (this is also well seen in Figure 9).

Typographic errors:
p.2 l.55: do you mean "tack" or task?
p. 7 l.125: correct "mensicus"
p. 10, l.179: "Table 1 list" -> Table 1 lists

References:
As previously stated, you need to complete the reference section
Apart from this, Ref 6: title is in capital letters.

Round 2

Reviewer 1 Report

The authors have nicely tackled all suggested changes, I think the paper is now much more detailed and does not lack anymore details of the hypothesis for the numerical simulations. I can thus recommend it in its present form for publication in Fluids.